# Integrated evaluation of telomerase activation and telomere maintenance across cancer cell lines

Kevin Hu[1,2,3], Mahmoud Ghandi[1†‡], Franklin W Huang[1,2,3]*

[1]Broad Institute of MIT and Harvard, Cambridge, United States; [2]Division of Hematology/Oncology, Department of Medicine; Bakar Computational Health Sciences Institute; Institute for Human Genetics; University of California, San Francisco, San Francisco, United States; [3]Helen Diller Family Comprehensive Cancer Center, San Francisco, United States

**Abstract** In cancer, telomere maintenance is critical for the development of replicative immortality. Using genome sequences from the Cancer Cell Line Encyclopedia and Genomics of Drug Sensitivity in Cancer Project, we calculated telomere content across 1299 cancer cell lines. We find that telomerase reverse transcriptase (*TERT*) expression correlates with telomere content in lung, central nervous system, and leukemia cell lines. Using CRISPR/Cas9 screening data, we show that lower telomeric content is associated with dependency of CST telomere maintenance genes. Increased dependencies of shelterin members are associated with wild-type *TP53* status. Investigating the epigenetic regulation of *TERT*, we find widespread allele-specific expression in promoter-wildtype contexts. *TERT* promoter-mutant cell lines exhibit hypomethylation at PRC2-repressed regions, suggesting a cooperative global epigenetic state in the reactivation of telomerase. By incorporating telomere content with genomic features across comprehensively characterized cell lines, we provide further insights into the role of telomere regulation in cancer immortality.

*For correspondence:
franklin.huang@ucsf.edu

Present address: †Monte Rosa Therapeutics, Boston, United States; ‡Cambridge Data Science LLC, Belmont, United States

## Introduction

Telomeres, repetitive nucleoprotein complexes located at chromosomal ends, are an important component of genomic stability (*de Lange, 2009*). As protective chromosomal caps, telomeres prevent potentially lethal end-fusion events (*McClintock, 1941*; *McClintock, 1942*) and mis-processing of chromosomal ends as damaged sites by the DNA repair machinery (*de Lange, 2005*; *Verdun and Karlseder, 2007*). Due to factors such as incomplete DNA replication and oxidative stress, telomeres gradually shorten with successive rounds of cell division (*Olovnikov, 1973*). If left unchecked, telomere attrition eventually triggers growth arrest and senescence, and further shortening can lead to acute chromosomal breakage and cell death. Unrestricted telomere shortening therefore acts as a major obstacle in the course of tumor development (*Hackett and Greider, 2002*; *Lorbeer and Hockemeyer, 2020*), and inhibition of telomere maintenance offers still largely untapped opportunities for targeted cancer therapies (*Damm et al., 2001*; *Dikmen et al., 2005*; *Flynn et al., 2015*).

Telomere shortening in embryonic development and in certain adult cell populations is offset by telomerase (*Greider and Blackburn, 1985*), a ribonucleoprotein enzyme with a core reverse transcriptase, *TERT,* that lengthens telomeres by catalyzing the addition of TTAGGG nucleotide repeats from an inbuilt RNA template component, *TERC* (*Feng et al., 1995*; *Yu et al., 1990*). Although telomerase is transcriptionally silenced in the majority of somatic cells, telomerase is reactivated in over 85% of all human cancers (*Kim et al., 1994*). Despite having activated telomere maintenance mechanisms, most cancers tend to have shorter telomeres than normal tissues, perhaps due to telomere

maintenance mechanisms developing only after a critical state of telomere crisis has been reached (*Okamoto and Seimiya, 2019*). Reactivation of telomerase is associated with a diverse set of genomic alterations, the most common of which include highly recurrent mutations in the *TERT* promoter (*Horn et al., 2013*; *Huang et al., 2013*), aberrant methylation (*Lee et al., 2019*) or copy number amplification (*Zhang et al., 2000*) of *TERT*, and modulation of the numerous transcription factors that regulate *TERT* expression (*Greider, 2012*; *Wu et al., 1999*). Of the minority of cancers that do not reactivate telomerase, many depend upon alternative lengthening of telomeres (ALT), a process that exploits mechanisms of homologous recombination and is characterized by heterogeneous telomere lengths, mutations in the *ATRX* and *DAXX* chromatin-regulating factors, and genome instability (*Cesare and Reddel, 2010*).

The readily identifiable nature of telomeric DNA repeats has motivated the development of computational methods for the determination of telomere content from whole-genome sequencing (WGS) and whole-exome sequencing (WES) data (*UK10K Consortium et al., 2014*). Recently, such methods were employed to characterize telomere content across tumor sequencing data from panels such as The Cancer Genome Atlas (TCGA), the Genotype-Tissue Expression (GTEx) project, and the Pan-Cancer Analysis of Whole Genomes (PCAWG) study, which have identified genomic markers of relative telomere lengthening and maintenance mechanisms (*Barthel et al., 2017*; *Castel et al., 2019*; *PCAWG-Structural Variation Working Group et al., 2020*). To gain a greater functional understanding of the landscape of telomere maintenance in cancer, we estimated telomeric DNA content (subsequently referred to as telomere content *Feuerbach et al., 2019*) across a diverse array of human cancer cell lines profiled in the Cancer Cell Line Encyclopedia (CCLE) (*Barretina et al., 2012*; *Ghandi et al., 2019*) and Genomics of Drug Sensitivity in Cancer (GDSC) (*Yang et al., 2013*) projects. Although cancer cell lines are immortalized, often through TERT activation, cell lines can serve as models for living tissues in aspects such as gene expression and have been deeply profiled (*Barretina et al., 2012*; *Ghandi et al., 2019*). We hypothesized that assessing telomere content could reflect underlying mechanisms of attrition, maintenance, and repair, which may be reflected in associations with genetic markers. By combining these estimates with a rich set of existing CCLE annotations, we aimed to determine genetic, epigenetic, and functional markers of telomere content and telomerase activity across a diverse panel of human cancer cell lines.

## Results

### Telomere content across cancer cell lines

Telomeric reads can be identified in DNA-sequencing reads using the canonical tandemly repeated TTAGGG motif, and normalized telomeric read counts may provide an accurate estimate of telomere content (*UK10K Consortium et al., 2014*). We quantified telomere content across cell lines using WGS and WES data from the independent CCLE (*Barretina et al., 2012*; *Ghandi et al., 2019*) and GDSC (*Yang et al., 2013*) datasets. In particular, we considered 329 cell lines profiled with WGS and 326 with WES in the CCLE, and 1056 samples profiled with WES in the GDSC, of which 55 were non-cancerous matched-normal samples. We note that our estimates state telomeric DNA repeat tract content, which is a normalized measure of telomeric reads in a sample, rather than solely telomere length, because telomere length requires the identification of true telomeric DNA from intrachromosomal, non-terminal telomeric DNA repeat tracts and extrachromosomal telomeric DNA (*Feuerbach et al., 2019*). To assess the fidelity of our telomere content measurements, we examined the agreement between the telomere content estimates in overlapping cell lines from independent sequencing datasets (*Figure 1—figure supplements 1* and *2a*). We observed high agreement between the telomere content estimates derived from CCLE WGS and GDSC WES data ($r = 0.84$, p $= 3.7 \times 10^{-79}$, $n = 286$) and moderate agreement between CCLE WGS and CCLE WES estimates ($r = 0.71$, p $= 1.5 \times 10^{-6}$, $n = 36$), suggesting that our length estimates were not strongly biased by source. Among the three datasets used, CCLE WGS samples each captured at least ten thousand telomeric reads (which we estimated as those with six or more TTAGGG repeats), followed by GDSC WES samples at 100–1000 telomeric reads, and lastly by CCLE WES samples with 10–100 telomeric reads (*Figure 1—figure supplement 2b*). Moreover, one cell line (HEL 92.1.7) was twice-sequenced in the CCLE WGS dataset and these replicates had similar raw telomere length estimates (4.00 and 4.06 kilobases). Based on the agreement between CCLE WGS and GDSC WES, we generated a

merged telomere content dataset of 1099 cell lines (Supplementary Methods) by combining the normalized log-transformed telomere contents derived from the CCLE WGS and GDSC WES datasets for downstream analyses.

Given these telomere content estimates, we first sought to examine the general distribution across cell lines and with respect to key cell line attributes such as donor age and population doubling rate. The overall distribution of telomere content displayed a slight skew (*Figure 1—figure supplement 2a*) toward longer telomeres, perhaps reflective of cell lines dependent upon ALT, a hallmark of which is telomeres of abnormal and heterogeneous lengths (*Bryan et al., 1997*; *Heaphy et al., 2011b*). We matched a substantial number of cell lines (282 for CCLE WGS, 554 for GDSC WES) with the age of the donor at the time of removal, from which we observed weak negative (vs. CCLE WGS: $r = -0.05$, p = 0.39; vs. GDSC WES: $r = -0.17$, p = $6.0\times10^{-5}$) correlations between telomere content and the age of the original donor (*Figure 1—figure supplement 3a*). Similarly, we also found TC to be negatively correlated with the log CCLE-calculated cell line doubling rate (vs. CCLE WGS: $r = -0.17$, p = 0.01; vs. GDSC WES: $r = -0.11$, p = 0.03) (*Figure 1—figure supplement 3b*), consistent with shorter telomeres in cell lines modestly associated with higher doubling rates. Among 1099 merged CCLE WGS and GDSC WES samples, we found raw telomere content to vary substantially both between (p = $2.0\times10^{-15}$, Kruskal-Wallis *H* test) and within (*Figure 1*) cell lines of different primary sites (*Figure 1—figure supplement 3c,d*). Cell lines of hematopoietic origin (namely leukemias and lymphomas, which comprised 156 lines) tended to have higher telomere contents on average (p = $2.0\times10^{-8}$, two-sided Mann-Whitney *U* test), perhaps due to their elevated levels of telomerase expression, which were the highest among all subtypes. The 55 non-cancerous samples profiled as part of the GDSC displayed relatively high telomere contents (p = $5.9\times10^{-4}$, two-sided Mann-Whitney *U* test, *Figure 1*), consistent with previous reports of widespread telomere shortening in cancer (*Barthel et al., 2017*). The greatest median telomere content, however, was found across lymphocyte and blood cell lines (*Figure 1*). The cell line with the highest telomere content was the U2-OS osteosarcoma line, a well-characterized model for ALT (*Bryan et al., 1997*).

## Genomic alterations associated with telomere content

Having determined telomere content in the context of cell line meta-attributes, we next sought to leverage the substantial genomic profiles of the CCLE to find correlates of telomere content. To characterize the genomic signatures of telomere content, we correlated the CCLE WGS and GDSC WES telomere content estimates against molecular annotations in the CCLE, with a focus on alterations known to be associated with telomere maintenance. First, we observed that telomere content and *TERT* overall and isoform-specific mRNA levels were positively though weakly correlated as determined by RNA sequencing (RNA-seq) both within each subtype (*Figure 1—figure supplement 4a,b*) and overall (*Figure 1—figure supplement 4c,d*). In specific cancer types such as CNS, lung, and leukemia, we found a higher correlation between *TERT* mRNA expression and telomere content (*Figure 1—figure supplement 4a,b*). Furthermore, we found negative associations between *TERT* mRNA levels and telomere content in bone and peripheral nervous system cell lines (*Figure 1—figure supplement 4a,b*). Because ALT is most commonly found in these cancer types, this may be a consequence of the near-mutual exclusivity between *TERT* expression and markers of ALT (*Killela et al., 2013*; *Lee et al., 2018*). Although mutations in *ATRX* and *DAXX* are closely associated with the development of the ALT phenotype (*Brosnan-Cashman et al., 2018*; *Clynes et al., 2015*; *Heaphy et al., 2011a*; *ALT Starr Cancer Consortium et al., 2012*; *Ramamoorthy and Smith, 2015*), comparisons between telomere content and mutations in *ATRX* and *DAXX* yielded significant associations only between *DAXX* alterations and merged telomere content (*Figure 1—figure supplement 5a*). We further repeated association tests with *TP53*, *VHL*, and *IDH1* as identified previously among TCGA samples (*Barthel et al., 2017*) and confirm that truncating *VHL* mutations are associated with reduced telomere lengths (*Figure 1—figure supplement 5a*), although this may be confounded by the high occurrence of *VHL* mutations in kidney cell lines. Testing telomere content association with molecular features profiled in the CCLE, we identified several genes known to be associated with telomere biology, suggesting that our integration of telomere content with existing annotations could identify features relevant to telomere maintenance mechanisms (*Figure 1—figure supplement 5b*). While we found relatively few significant associations between telomere content and mutations, we note that we were limited to an absolute estimate of telomere content as

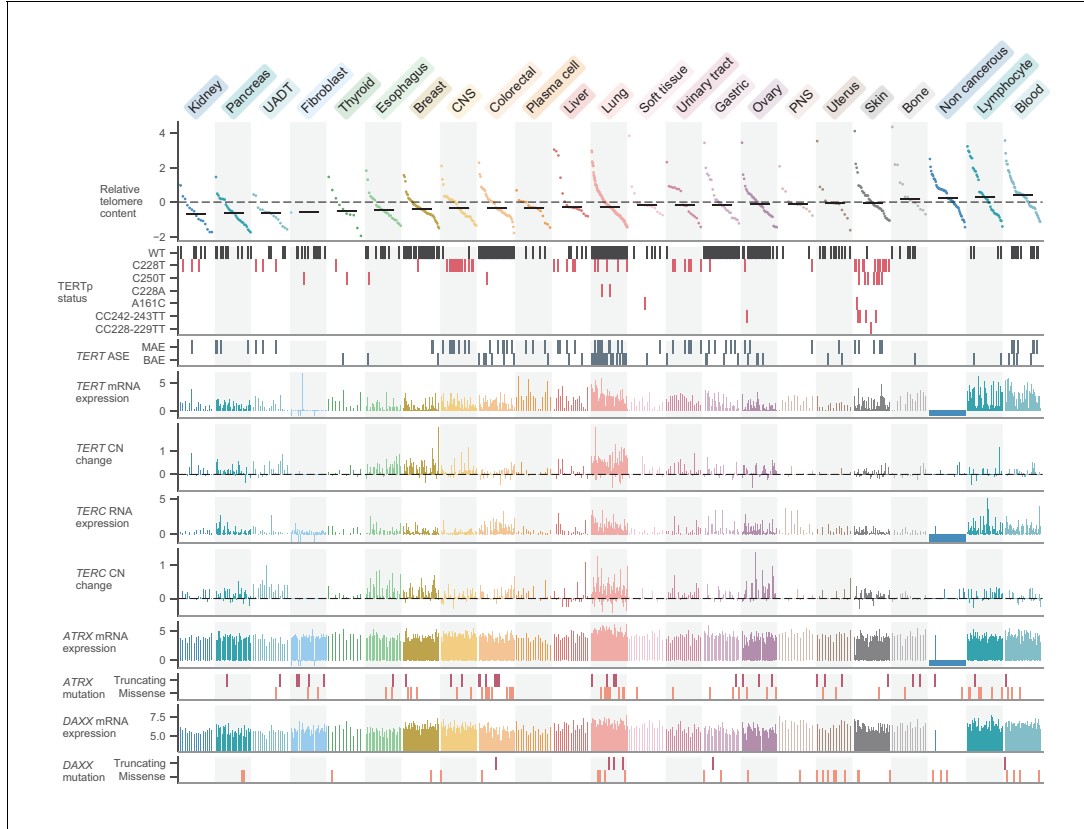

**Figure 1.** Telomere content and related genomic features across human cell lines. Cell lines were grouped by cancer type and ordered by telomere content within each type, and are displayed such that each column represents a cell line. Telomere content measurements reflect combined z-scored estimates derived from CCLE WGS and GDSC WES with means for samples with telomere content estimates from both sources. Bars within each cancer type represent medians. Relative copy number values are shown as $\log_2$(relative to ploidy + 1)–1. Cell lines shown are filtered such in addition to annotations for telomere content, values for *TERT* and *TERC* RNA-seq expression, *TERT* and *TERC* copy number, and *ATRX* and *DAXX* mutation status are all available (with an exception made for non-cancerous cell lines, which lack such profiling in DepMap). Cell lines were also filtered such that each cancer type is represented by at least 10 cell lines (n = 738 cell lines total). RNA expression estimates are in terms of $\log_2$(TPM+1). CNS: central nervous system; PNS, peripheral nervous system; UADT, upper aerodigestive tract.

The online version of this article includes the following figure supplement(s) for figure 1:

**Figure supplement 1.** Overlap between cell lines represented in annotations.

**Figure supplement 2.** Telomere content agreement between sequencing sets.

**Figure supplement 3.** Telomere content, age, and tissue subtype.

**Figure supplement 4.** Transcriptomic associations between *TERT*, *TERC*, and telomere content.

**Figure supplement 5.** Associations between telomere content and cell line characteristics.

opposed to a relative measure of somatic telomere lengthening, which requires a paired normal sample (*Barthel et al., 2017*).

## Telomere content associates with CST complex dependencies

Given the associations between telomere content and several genomic and transcriptomic features, we next considered whether variations in telomere content could confer or reduce selective vulnerabilities to inactivation of certain genes. In particular, we hypothesized that telomere content may be associated with vulnerabilities to reductions in the levels of telomere-regulating proteins. These vulnerabilities may be measured using CRISPR- and RNAi-based depletion assays, which can be processed to yield a numerical value for the dependency of a gene within a cell line, with more negative values indicative of increased dependence on a particular gene. To reveal such associations, we correlated our telomere content estimates with gene inactivation sensitivities assessed via genome-wide CRISPR-Cas9 (Avana; *Meyers et al., 2017*) and RNAi viability screens (Achilles RNAi

[*Tsherniak et al., 2017*] and DRIVE [*McDonald et al., 2017*]). Although we found no dependencies that displayed outlier associations with telomere content in the Achilles RNAi screen (*Figure 1—figure supplement 5b*), we discovered that sensitivity to Avana CRISPR-Cas9 knockouts of each of the three CST complex proteins as well as the telomere-associating protein TERF1 were outlier associations with telomere content estimates computed from both the GDSC WES and CCLE WGS data (*Figure 2a,b*). Specifically, sensitivity to knockout of the CST complex components (*CTC1, STN1, TEN1*), which are key mediators of telomere capping and elongation termination (*Chen et al., 2012*), was correlated with lower telomere content (*Figure 2—figure supplement 1a,b*). Although the CST complex was not assessed in the DRIVE screening dataset, we again found *TERF1*, a key shelterin component, to be among the positively correlated genes with telomere content in the

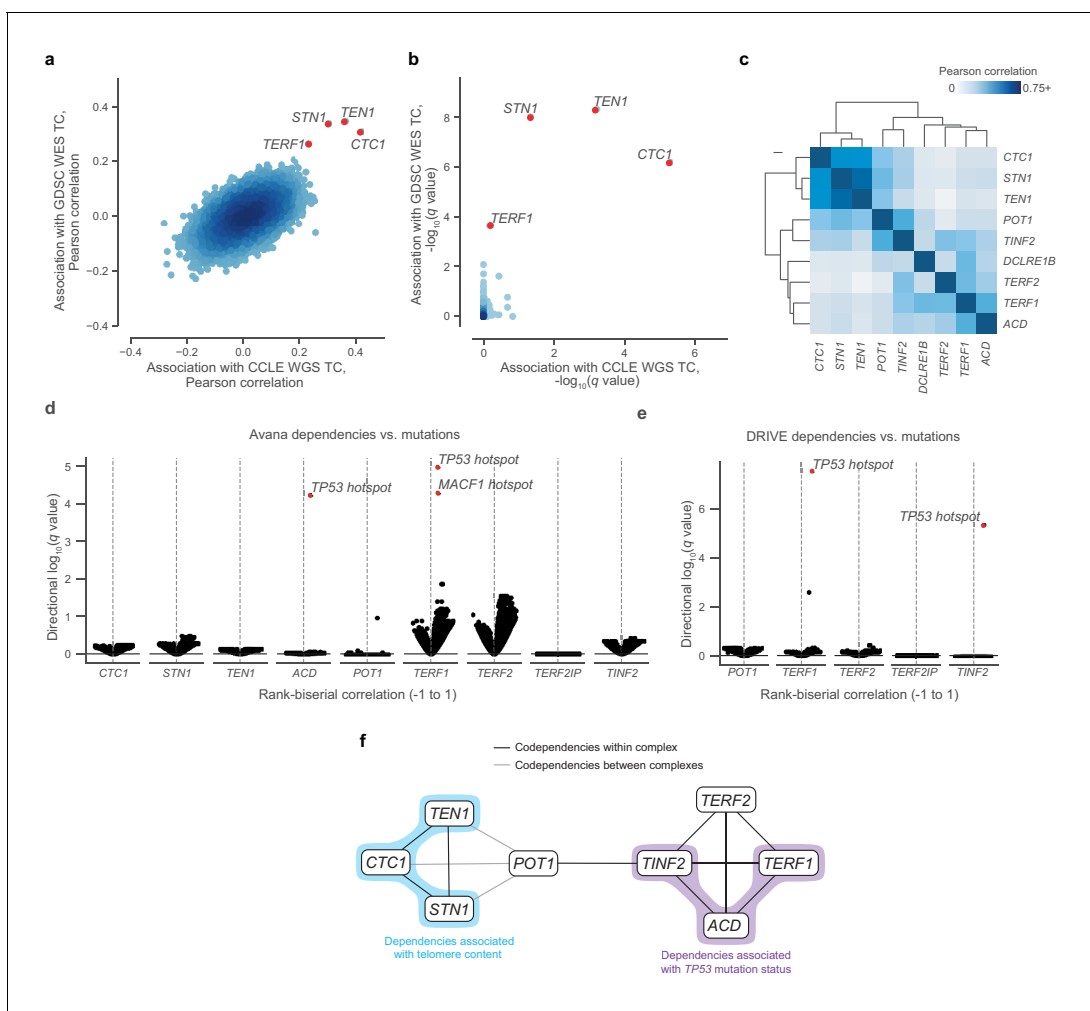

**Figure 2.** Telomere-binding protein dependencies are associated with telomere content and *TP53* mutation status. (a) Pairwise plot of Pearson correlations between dependencies of all genes in the Avana dataset and CCLE WGS telomere content (x-axis, n = 210–211 cell lines) and GDSC WES telomere content (y-axis, n = 420–426 cell lines) estimates. (b) Pairwise plot of significance levels of correlations shown in (a) with correction for multiple hypothesis testing. (c) Pairwise Pearson correlation matrix between Avana dependencies among CST members and five shelterin components (n = 796–808 cell lines; *Supplementary file 3*). (d) Associations of CST and shelterin member Avana dependency scores with damaging and hotspot mutations (n = 796–808 cell lines). For each gene dependency, mutation associations were computed using rank-biserial correlations with mutants and wild-types as the two categories. p Values determined using two-sided Mann-Whitney *U* test. (e) Associations of shelterin member DRIVE dependency scores with damaging and hotspot mutations (n = 372–375 cell lines; *Supplementary file 3*) under the same scheme used in (d). (f) Network schematic of the co-dependency matrix shown in (c) and annotated with association with telomere content or *TP53* mutation status.

The online version of this article includes the following figure supplement(s) for figure 2:

**Figure supplement 1.** Telomere content and telomere protein dependencies.

**Figure supplement 2.** *TP53* mutation status and shelterin member dependencies.

DRIVE panel (*Supplementary file 2*). Overall, these data suggest that cancer cell lines with shorter telomeres are more susceptible to inhibition of the CST-mediated telomere maintenance mechanism.

Using the CST complex as a seed set, we subsequently queried all dependencies under the premise that associated gene dependencies reflect coordinated functions (*Pan et al., 2018*). Within the Avana panel (*n* = 757–769), we found significant (FDR < 0.01) outlier associations between the CST complex genes and genes encoding six additional telomere-associating proteins (*ACD, POT1, TERF1, TERF2, TINF2,* and *DCLRE1B*). These first five additional telomere-associating proteins, together with *TERF2IP*, comprise the shelterin complex, the protector and regulator of telomere length and topology (*de Lange, 2005*). Interestingly, whereas the five other shelterin dependencies were positively associated with telomere content, *TERF2IP* displayed a weak negative association (*Figure 2—figure supplement 1c*), suggesting that *TERF2IP* may play a distinct regulatory role in shelterin function compared to the other members. To examine the dependency landscape of the CST complex and these six other telomere-related proteins, we computed a correlation matrix involving these nine genes, clustering of which yielded two main subgroups: one comprised of the CST complex members, and another of the six other genes (*Figure 2c*). Despite this separation, *POT1* and *TINF2* also displayed notable correlations with CST dependencies, possibly serving as the primary mediators of previously-reported functional interactions between the shelterin and CST complexes (*Chen et al., 2012*; *Wan et al., 2009*).

Although we found strong codependency relationships within this group of telomere-associated proteins, we also observed that certain shelterin members displayed notable codependencies with p53 pathway members such as *MDM2, ATM,* and *TP53* itself (*Figure 2—figure supplement 2a,b*). Because sensitivity to perturbation of the p53 pathway is highly associated with *TP53* mutations in cancer (*McDonald et al., 2017*), we asked if these codependency relationships were also associated with hotspot mutations in *TP53*. In fact, *TP53* was a significant (*FDR* < 0.001) outlier when a comprehensive set of hotspot and damaging mutations was compared against sensitivity to *ACD* and *TERF1* dependencies in the Avana panel (*Figure 2d*, *Figure 2—figure supplement 2c*) and against *TERF1* and *TINF2* dependencies in the DRIVE panel (*Figure 2e*, *Figure 2—figure supplement 2d*). These links between these gene dependencies and *TP53* mutation status reprise and extend previous reports of p53-dependent DNA damage responses to *TERF1* and *TINF2* depletion (*Pereboeva et al., 2016*; *Rosenfeld et al., 2009*). Taken together, we find that CST and shelterin dependencies are correlated with each other, telomere content, and *TP53* mutation status (*Figure 2f*).

## Patterns and mechanisms of telomerase expression

Having thoroughly characterized telomere content and its related dependencies across the CCLE, we next focused on the regulation of *TERT* transcription itself. Across 1019 samples previously profiled with deep RNAseq, we found that hematopoietic cell lines (leukemias, lymphomas, and myelomas) were associated with the greatest mean expression of *TERT* (p = $5.8 \times 10^{-26}$, two-sided Mann-Whitney *U* test; *Figure 1*). In contrast, *TERT* expression was significantly reduced (p = $2.0 \times 10^{-22}$, two-sided Mann-Whitney *U* test) and generally undetectable in fibroblast-like cell lines. With regard to the telomerase RNA component (*TERC*), we found strong associations between *TERC* RNA expression and RNA levels of several small Cajal body-specific RNAs (scaRNAs) and histone subunit RNAs (*Figure 1—figure supplement 4e*). The co-expression of *TERC* and these other RNAs may be a consequence of their shared localization, processing, and regulation in Cajal bodies (*Gall, 2003*; *Venteicher et al., 2009*; *Zhu et al., 2004*), as *TERC* itself contains an H/ACA box small nucleolar RNA (snoRNA) domain (*Mitchell et al., 1999*) and is a scaRNA family member. Moreover, these associations suggest that variations in Cajal body processing may act as factors in *TERC* reactivation (*Cao et al., 2008*). Although we focused on the transcriptional features of telomerase, it is important to note that other factors in addition to *TERT* and *TERC* expression determine telomerase activity and the eventual maintenance of telomere length (*Listerman et al., 2013*). Despite these orthogonal factors, *TERT* enzymatic activity remains strongly correlated with raw levels of *TERT* expression (*Takakura et al., 1998*).

Beyond raw gene and transcript expression levels, we next examined the underlying mechanisms for reactivating TERT in cancer using the comprehensive cell line data. Widespread transcriptional reactivation of *TERT* in cancer is driven by a variety of factors. Aside from copy number

amplifications (*Zhang et al., 2000*), highly recurrent mutations in the *TERT* promoter drive strong monoallelic *TERT* re-expression (*Bell et al., 2015*; *Huang et al., 2015*). To explore the intersections between methylation, promoter mutations, and allele-specific expression, we combined available profiling data from the CCLE (*Figure 3a*). First, using WGS and targeted sequencing of the *TERT* promoter provided in the CCLE, we assessed *TERT* promoter status for 503 cell lines across 21 cancer types (*Figure 3—figure supplement 1a*). We found that only the C228T (chr5:1,295,228 C>T) mutation was significantly (p = $2.8 \times 10^{-5}$, two-sided Mann-Whitney *U* test) associated with an increase in TERT expression (*Figure 3—figure supplement 1b*). Surprisingly, the mean level of *TERT* expression in monoallelic contexts was only slightly lower than that of biallelic contexts, with less

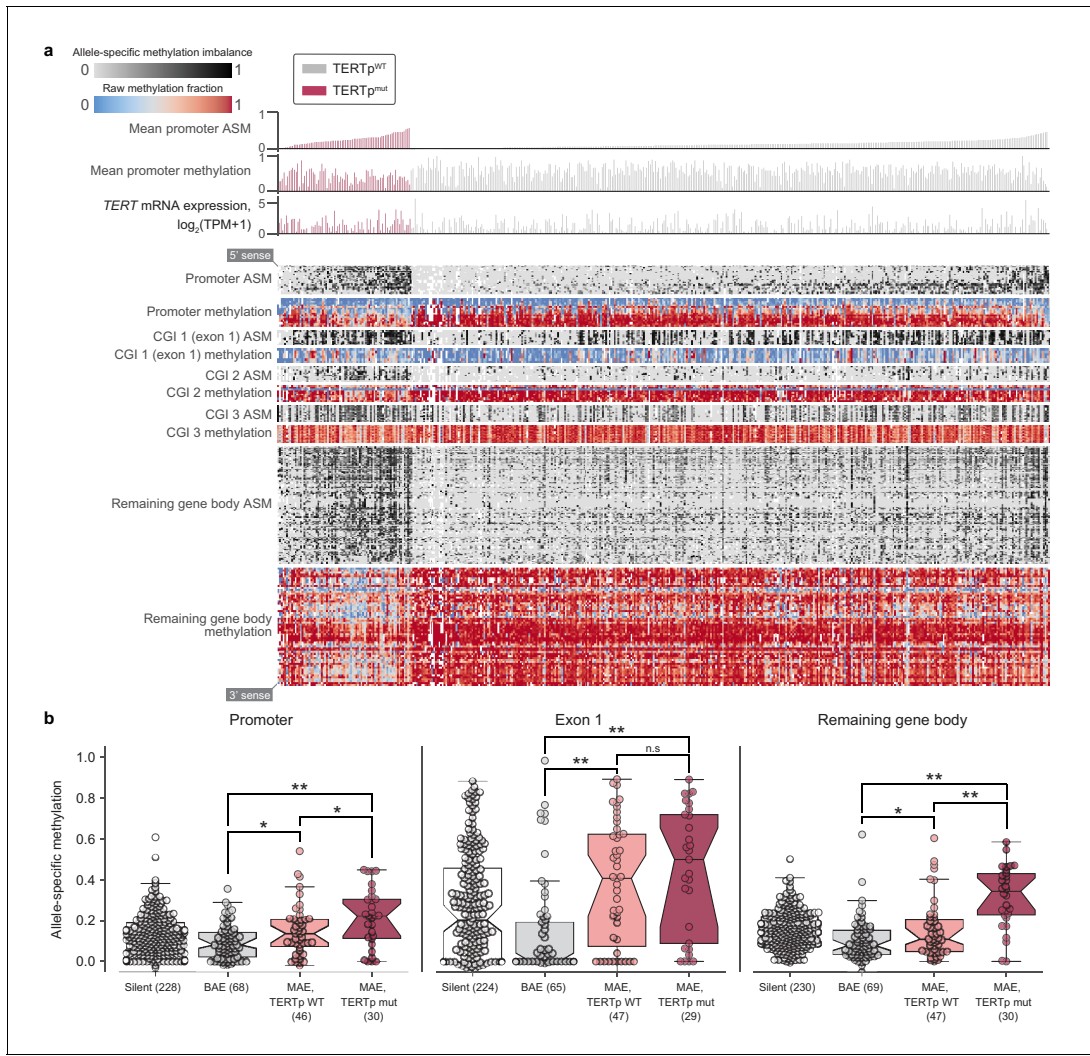

**Figure 3.** Allele-specific methylation of the *TERT* locus is indicative of both promoter mutation status and allele-specific expression. (**a**) Heatmap of CpG methylation levels along the *TERT* locus, sorted in order of mean methylation levels along the upstream 5 kb region within TERTp-mutants and -wildtypes. *TERT* gene expression levels are also indicated for each cell line. Each column represents a cell line (*n* = 450), and each row represents a CpG pair (*n* = 209) sorted from the 5' to 3' direction along the *TERT* sense strand. White blocks indicate missing ASM/methylation values. Cell lines with unavailable ASM values for at least half of *TERT* locus CpGs were excluded. (**b**) ASM levels of *TERT* locus subregions in cell lines are indicative of TERTp status and allele-specific expression. BAE, biallelic expression; MAE, monoallelic expression. Boxes, interquartile range (IQR); center lines, median; whiskers, maximum and minimum or 1.5 × IQR; notches, 95% confidence interval of bootstrapped median using 1000 samples and a Gaussian-based asymptotic approximation. *p < 0.05, **p < 0.01, n.s, not significant; two-sided Mann-Whitney *U* test.

The online version of this article includes the following figure supplement(s) for figure 3:

**Figure supplement 1.** *TERT* promoter mutations, gene expression, and ASE.

**Figure supplement 2.** Interactions bewteen *TERT* ASE, promoter mutations, and methylation.

than a 1.5-fold difference between the groups (p = 0.03, two-sided Mann-Whitney *U* test). Given that cells with biallelic *TERT* expression presumably express *TERT* with twice the transcriptional source sites as those with monoallelic *TERT* expression, this reduced difference may be a consequence of the effects of the *TERT* promoter mutation in producing particularly robust monoallelic expression (*Huang et al., 2013*) or expression of TERT from multiple sites all of the same allele (*Rowland et al., 2019*).

To further explore allele-specific expression (ASE) patterns of *TERT*, we employed an ASE-calling pipeline (Supplementary Methods) and determined *TERT* allele-specific expression status for 157 cell lines (*Figure 3—figure supplement 1c*), an increase of 69 cell lines compared to a previous report using CCLE WGS data (*Huang et al., 2015*). Out of these 157 cell lines, 87 (58.6%) express *TERT* from a single allele. Moreover, of these 157 cell lines, 129 have a sequenced promoter, with which we confirm that promoter mutations unanimously drive monoallelic expression (*Figure 3—figure supplement 2a*). Our expanded set of cell lines also reveals several new tissues of origin in which *TERT* is monoallelically expressed without a mutant promoter, such as hematopoietic cell lines (*Figure 3—figure supplement 2b*). This high proportion of *TERT* monoallelic expression led us to ask whether there are genomic alterations aside from promoter mutations that could lead to ASE. Under the assumption that such alterations may also induce ASE in a larger region than a promoter mutation, we determined ASE status in *SLC6A19* and *CLPTM1L*, which are the most immediate neighbor genes of *TERT*. Because the number of samples with annotated ASE in both *TERT* and these neighbors was not large enough for comparisons of overlapping ASE, we instead examined the individual frequencies of ASE among these three genes. Compared to promoter-wildtype monoallelic *TERT* expression occurring in 36% (47 of 129) of samples, only 9.4% (11 out of 117) of samples expressed *CLPTM1L* from a single allele and 22% (5 of 22) expressed *SLC6A19* from a single allele, and we were unable to assess co-occurrence due to lack of overlap in samples with heterozygous SNPs in both *TERT* and *CLPTM1L* or *SLC6A19*. Furthermore, a search for structural variants in the surrounding 100 kilobase regions yielded no significant associations. However, given that only 106 cell lines had both a known *TERT* ASE status and the required WGS data for structural variant determination, more genomic annotations may be needed for the discovery of additional mechanisms driving the monoallelic expression of *TERT*.

## Distinct *TERT*p methylation patterns at the *TERT* locus

Aside from monoallelic expression, *TERT* promoter mutations are characterized by unique patterns of epigenetic marks, namely allele-specific CpG methylation (ASM) and H3K27me3 repressive histone modifications (*Stern et al., 2015*; *Stern et al., 2017*) and long-range chromatin interactions (*Akıncılar et al., 2016*). Using genome-wide RRBS data across 928 cell lines, we elucidated associations between CpG-site methylation of the *TERT* locus (namely, *TERT* and the surrounding 5 kb) and *TERT* promoter mutations. Examination of methylation patterns at the *TERT* locus revealed five prominent ASM clusters in the *TERT* locus, corresponding roughly to the upstream 5 kb region (containing the promoter), part of a CpG island overlapping the first exon, two other parts of this CpG island, and the remaining gene body (*Figure 3a*). Comparison of each region's mean ASM against TERTp mutant status revealed that TERTp mutants exhibited strong and significant (p < 0.01) increases in ASM in the promoter (*n* = 485), remaining gene body (*n* = 493), and exon 1 (*n* = 478) regions (*Figure 3b*, *Figure 3—figure supplement 2c*). In contrast, TERTp wild-type cell lines tended to lack ASM throughout the *TERT* locus, instead being hypermethylated in all regions except for exon 1 (*Figure 3b*, *Figure 3—figure supplement 2d*), and partial hypomethylation in promoter mutants may reflect the hemizygous methylation previously observed at the *TERT* locus (*Rowland et al., 2020*; *Stern et al., 2015*; *Stern et al., 2017*). Absolute methylation of the remaining gene body was positively correlated with *TERT* expression in both promoter status contexts (*Figure 3—figure supplement 1e*), which parallels previous reports of a positive correlation between *TERT* expression and methylation (*Barthel et al., 2017*; *Salgado et al., 2019*). Exon 1 methylation was elevated in nearly all cell lines with monoallelic *TERT* expression in both the mutant promoter context (p = $6.3 \times 10^{-3}$, two-sided Mann-Whitney *U* test, *n* = 81) and the wildtype promoter context (p = $1.6 \times 10^{-4}$, two-sided Mann-Whitney *U* test, *n* = 98) compared to biallelic *TERT* expressors (*Stern et al., 2017*). Interestingly, although most cell lines with monoallelic *TERT* expression displayed partially elevated methylation levels in exon 1 (*Figure 3b*), only promoter-mutant cell lines

were hypomethylated in the surrounding regions, suggesting that the epigenetic state of promoter mutants is in fact distinct from that of promoter-wildtype monoallelic *TERT* expressors.

## A genome-wide epigenetic pattern in TERTp mutants

The observation that *TERT* promoter mutants display a hypomethylated *TERT* locus even compared to other monoallelic *TERT* expressors led us to ask if additional epigenetic signals are indicative of *TERT* promoter status. In particular, we considered the possibility that epigenetic changes to the *TERT* locus could in fact act as a cooperative factor (*Kim et al., 2016*; *Kim and Shay, 2018*) in tumorigenesis or tumor cell maintenance rather than as a consequence of *TERT* promoter mutations. To address this hypothesis, we performed a genome-wide search for CpG islands (CGIs) with significant differences in methylation levels in *TERT*p mutant cell lines compared to *TERT*p wild-type ones. If *TERT* hypomethylation were a downstream consequence of *TERT* promoter mutations then we would expect *TERT* hypomethylation to be an isolated event, and thus there would be few CGIs outside the vicinity of *TERT* with methylation levels correlated with TERTp mutant status. Surprisingly, we instead found a broad genome-wide distribution of CGIs that were hypomethylated in TERTp[mut]

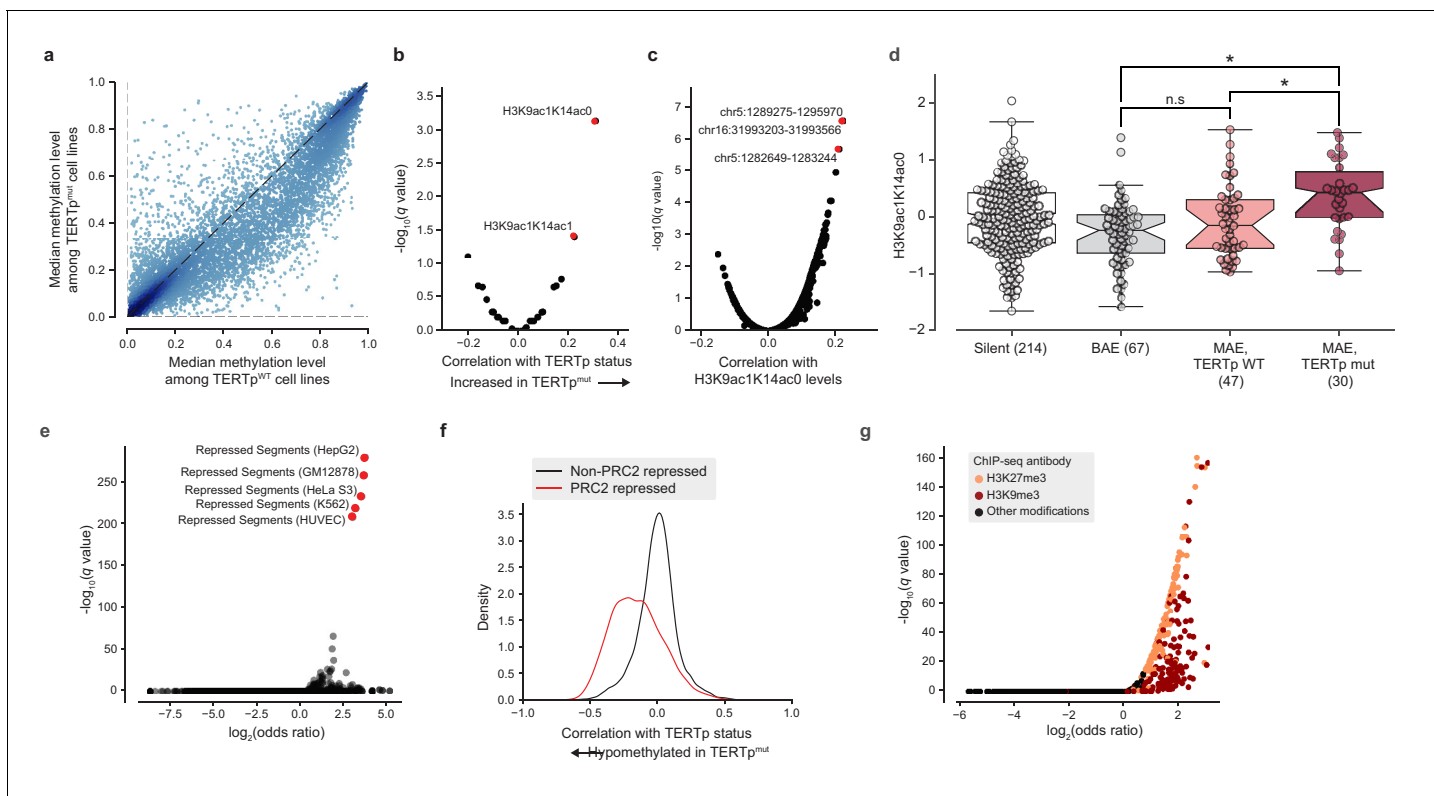

**Figure 4.** *TERT* promoter mutations associate with genome-wide decreased methylation of PRC2-repressed regions. (**a**) Pairwise plot of median CGI methylation levels in TERTp[mut] cell lines ($n = 21–83$; ***Supplementary file 6***) versus TERTp[WT] cell lines ($n = 95–410$, ***Supplementary file 6***). Each dot represents a CGI. (**b**) Rank-biserial correlations between TERTp status (mutant or wild-type) and global histone modification levels ($n = 302–475$). Significance determined by two-sided Mann-Whitney $U$ test. (**c**) Pearson correlation levels between global H3K9ac1K14ac0 levels and ASM imbalance of CGIs ($n = 261–884$). (**d**) H3K9ac1K14ac0 levels are significantly increased in TERTp mutants. Boxes, interquartile range (IQR); center lines, median; whiskers, maximum and minimum or $1.5 \times$ IQR; notches, 95% confidence interval of bootstrapped median using 1000 samples and a Gaussian-based asymptotic approximation. *$p < 0.01$, n.s, not significant; two-sided Mann-Whitney $U$ test. (**e**) LOLA core set enrichment analysis of CGIs hypomethylated in TERTp[mut] cell lines reveals enrichment of PRC2-repressed regions. (**f**) Kernel density distributions of rank-biserial correlations between CGI methylation levels for PRC2-overlapping regions and non-PRC2-overlapping regions. A negative correlation indicates that a CGI is hypomethylated in TERTp[mut] cell lines relative to TERTp[WT] ones, and a positive correlation indicates the opposite. PRC2 regions were sourced from the HepG2 segmentation. (**g**) LOLA ENCODE Roadmap region enrichment analysis of CGIs hypomethylated in TERTp[mut] cell lines reveals enrichment of H3K9me3 and H3K27me3 regions.

The online version of this article includes the following figure supplement(s) for figure 4:

**Figure supplement 1.** Global methylation changes associated with *TERT* promoter mutations.

samples relative to $TERTp^{WT}$ samples (*Figure 4a*). Moreover, when correlated with a panel of global histone modification levels, we found that *TERT*p mutants exhibited increased levels of H3K9ac1K14ac0 and H3K9ac1K14ac1 marks (*Figure 4b*), which have been suggested as marks of transcriptionally active chromatin (*Ruthenburg et al., 2007*). Likewise, when H3K9ac1K14ac0 levels were compared against a genome-wide panel of CGI ASM levels, the *TERT* CGI (chr5:1,289,275–1,295,970) was the top correlate (*Figure 4c*). H3K9ac1K14ac0 levels were significantly increased in *TERT*p mutants compared to monoallelic *TERT*p wild-type cell lines, linking this histone modification to *TERT*p mutation (*Figure 4d*).

To better understand the distribution of these $TERTp^{mut}$-hypomethylated CGIs, we utilized Locus Overlap Analysis (LOLA) (*Nagraj et al., 2018*; *Sheffield and Bock, 2016*) to query the significance of overlaps between these CGIs and predetermined region sets. Among the top 1000 $TERTp^{mut}$-hypomethylated CGIs (*Figure 4a*), we found significant (*FDR* < 0.0001) and robust 10-fold enrichment for polycomb repressive complex 2 (PRC2)-repressed regions (*Figure 4e*) previously characterized in several cell lines (HepG2, GM12878, HeLa-S3, K562, and HUVEC). Beyond these top 1000 hypomethylated CGIs, CGIs overlapping with PRC2-repressed segments were broadly hypomethylated in $TERTp^{mut}$ cell lines and accounted for nearly all the previously observed skew toward hypomethylation (*Figure 4f*). Interestingly, the enrichment of PRC2 segments was much smaller (around 3.5-fold) in the remaining profiled cell line, H1-hESC. Against ENCODE ChIP-seq peak region sets, we also found significant overlap with the H3K9me3 and H3K27me3 heterochromatin marks (*Figure 4g*). Furthermore, we also observed a moderate twofold ($P = 5.1\times10^{-24}$, Fisher's exact test; *Figure 4—figure supplement 1a*) enrichment for regions within 10 megabases of most telomeres, consistent with previous reports that PRC2-repressed and H3K27me3-marked regions are enriched in telomeric and subtelomeric regions (*Rosenfeld et al., 2009*). The enrichment of these hypomethylated regions among telomere-proximal regions may also be indicative of a recently-reported telomere position effect, which has been shown to affect the chromatin accessibility of the *TERT* locus (*Kim et al., 2016*) located close to the chromosome 5 p telomere. Given that PRC2 has previously been shown to exhibit allele-specific binding to the methylated silent allele in $TERTp^{mut}$ cell lines (*Stern et al., 2017*), this genome-wide pattern of hypomethylation at PRC2 sites suggests that background epigenetic events may interact with promoter mutations in facilitating *TERT* expression.

Given this peculiar pattern of hypomethylation at telomere-proximal sites of PRC2 repression across CCLE samples, we next asked if a similar pattern exists across TCGA primary tumor samples. To test this hypothesis, we estimated CGI methylation levels across 878 TERT-expressing TCGA samples characterized with both the Illumina 450 k array and with a previously determined *TERT* promoter status (*Barthel et al., 2017*). Consistent with previous analyses of *TERT* methylation levels at the cg11625005 methylation probe, we find that $TERTp^{mut}$ samples tended to exhibit hypomethylation (*Figure 4—figure supplement 1b,c*). Among 13,547 CGIs, we again found an enrichment of hypomethylation of PRC2-overlapping CGIs (*Figure 4—figure supplement 1d*), although this was less prominent than previously noted in the CCLE. LOLA enrichment analysis for $TERTp^{mut}$-hypomethylated CGIs in the TCGA likewise confirmed significant enrichments (*FDR* < 0.0001) of PRC2-repressed regions and associated histone modifications as the top enriched region sets (*Figure 4—figure supplement 1e,f*). However, the fold-enrichment was less (about five-fold) than that observed in the CCLE and did not display any significant enrichment in telomere-proximal regions (p = 0.70, Fisher's exact test). Although the skew toward hypomethylation in *TERT* promoter mutants among these TCGA samples was weaker than in the CCLE samples, this may be the result of the more heterogeneous nature of primary TCGA samples as well as the differences in coverage between the Illumina 450 k array and RRBS.

## Discussion

To investigate the nature of telomeres and their maintenance mechanisms in cancer, we applied a functional genomics approach toward understanding molecular relationships across cancer cell lines. We estimated relative telomere content across over a thousand cancer cell lines and thus provide a useful reference for further studies on cancer cell line characteristics that have not to date considered this feature. We show that cell line telomere content indeed varies with factors such as tissue type, *TERT* mRNA expression, and mutations in genes such as *DAXX* and *VHL*. Moreover, we discovered novel relationships between telomere content and dependencies of CST and shelterin complex

members, which was enabled by the high overlap between the cell lines profiled by our estimates and by several loss-of-function screens (*McDonald et al., 2017*; *Meyers et al., 2017*; *Tsherniak et al., 2017*).

Using these genome-wide gene dependency estimates, we found that increased sensitivity (*Meyers et al., 2017*) to depletion of CST complex members correlates with shorter telomeres, likely a consequence of the critical roles of the CST complex in both telomere protection and in terminating telomere elongation (*Chen et al., 2012*). Our findings raise the possibility that targeting the CST complex may preferentially affect cancer cells that harbor shorter telomeres, and telomere content may be used as a biomarker of drug response in tumors. Likewise, CST complex dependencies were positively associated with the dependencies of several shelterin complex components, reflecting their functional interactions. Among these shelterin complex members, we find that the responses to their depletion are highly dependent upon the presence of a wild-type *TP53* gene, with *TP53* mutants displaying reduced sensitivity to depletion of *ACD*, *TERF1*, and *TINF2*. Additional studies are required to validate these associations and to assess why only certain members of the shelterin complex show this *TP53*-dependent sensitivity effect.

In addition to telomere content, we also investigated the genomic landscape of telomere maintenance mechanisms, namely mechanisms of *TERT* reactivation, across cancer cell lines. The enrichment of *TERT* promoter mutations in certain tissues has inspired several explanations, and our findings in both the CCLE and TCGA suggest a specific epigenetic signature that may underlie this unique pathway of telomere maintenance. We found that in *TERT* promoter mutants, CpG islands were preferentially hypomethylated in PRC2-repressed regions located near telomeres, which may relate to previous reports of a long-range telomere position effect (*Kim et al., 2016*; *Yuan et al., 2019*) and of *TERT* expression necessitating specific chromatin states in promoter-wildtype and mutant samples (*Salgado et al., 2019*). Considering that normal tissues typically exhibit particularly low methylation of the *TERT* promoter (*Salgado et al., 2019*; *Stern et al., 2017*) and that PRC2 occupies the inactive allele in *TERT* promoter mutants (*Stern et al., 2017*), our genome-wide signature may relate to the latter part of the two-step mechanism proposed for TERTp mutation-driven telomerase upregulation (*Chiba et al., 2017*). Moreover, epigenetic mechanisms have been shown to produce synergistic effects with driver mutations in tumor evolution (*Tao et al., 2019*). Besides reflecting a direct cooperation with *TERT* expression, this signature raises the possibility that the 'memory' of short telomeres may be preserved through these telomere-proximal hypomethylated regions. It may also be indicative of the stemness of cell lines, which has been proposed as a major factor in the proliferative advantage of *TERT* promoter mutations (*Chiba et al., 2015*). Future studies will be necessary to elucidate the nature of this epigenomic signature, how it impacts the regulation of telomerase expression, and the complexities of *TERT* expression beyond binary measures of allele-specificity (*Rowland et al., 2019*). Furthermore, incorporation of telomere content into studies using cancer cell lines may help improve our understanding of sensitivities to drugs or genetic perturbations across cell lines.

Through our analysis, we show relevant markers of telomere-associated protein function, patterns of *TERT* reactivation across cancers, and epigenetic determinants of *TERT* promoter status. We detail various features of telomere regulation and dysfunction in cancer, and we provide a substantial addition of new features to a well-characterized set of cell lines. By doing so, we complement molecular studies of telomeres in parallel studies across the GTEx (*GTEx Consortium et al., 2020*), TCGA (*Barthel et al., 2017*), and PCAWG (*PCAWG-Structural Variation Working Group et al., 2020*) panels, providing a resource that will guide additional studies on the roles and functions of telomeres in cancer.

## Materials and methods

### Telomere content estimation

Telomere content estimates were computed using Telseq (*UK10K Consortium et al., 2014*) with the default settings. Telseq records the frequencies of reads containing various frequencies of the canonical TTAGGG telomeric repeat, and then normalizes this number of telomeric repeats using a GC-adjusted coverage estimate and the average chromosome length.

Telomere content was estimated for WGS and WES samples in the CCLE (*Ghandi et al., 2019*) as well as WES samples in the GSDC (*Yang et al., 2013*) using the default settings. When multiple read groups were present in a sample, telomere content was computed as a mean of the individual read group estimates weighted by the total read count per group. Whereas we found decent agreement between overlapping samples in CCLE WGS and GDSC WES, we found a comparatively weak correlation between both sets and the CCLE WES estimates (*Figure 1—figure supplement 2*). Therefore, we excluded the CCLE WES telomere content estimates from subsequent analyses.

In comparing the CCLE WGS and GDSC WES estimates, we also noticed a batch effect resulting in two clusters of GDSC WES estimates. To identify and correct this batch effect across all GDSC WES estimates, we observed that these batches were distinguished by frequencies of reads containing exactly 4, 5, and 6 telomeric motifs. We then ran a k-means clustering on these read frequencies to estimate the clusters across all GDSC WES samples, which were subsequently adjusted by re-centering the mean of one cluster (after applying a z-scored log-transformation) to match the mean of the other.

We also attempted to use Telseq to estimate telomeric repeat-containing RNA (TERRA) expression across 1019 RNA-seq samples from the CCLE. However, because the majority of these samples were found to contain little or no reads containing telomeric reads, TERRA capture was determined to be too low for any meaningful analysis.

Cell lines were annotated with sample descriptors from the CCLE data portal (Cell_lines_annotations_20181226.txt, https://portals.broadinstitute.org/ccle/data). Harmonized sample information, telomere content estimates, and other matched annotations are available in *Supplementary file 1*.

## Genomic and transcriptomic markers

We sourced mutations and copy number estimates from the DepMap download portal (https://depmap.org/portal/download/) under the public 19Q4 release (CCLE_mutations.csv and CCLE_gene_cn.csv, respectively). We used the mutation classifications detailed in the *Variant_annotation* column.

We also downloaded processed RNAseq estimates in the form of gene expression, transcript expression, and exon inclusion estimates from the CCLE data portal under the latest release (CCLE_RNAseq_rsem_genes_tpm_20180929.txt.gz, CCLE_RNAseq_rsem_transcripts_tpm_20180929.txt.gz, and CCLE_RNAseq_ExonUsageRatio_20180929.gct.gz, respectively). We also downloaded RPPA estimates (CCLE_RPPA_20181003.csv) and global chromatin profiling results (CCLE_GlobalChromatinProfiling_20181130.csv). Before performing subsequent analyses, we transformed transcript and gene expression TPMs by taking a $\log_2$-transform with a pseudocount of +1. We also excluded transcripts with a standard deviation in this $\log_2(TPM + 1)$ measure of less than 0.25 across all cell lines. We excluded exons with missing inclusion values in over 800 cell lines or with a standard deviation of less than 0.1. Pearson correlations were then used to calculate associations between gene and transcript RNA expression levels of *TERT* and *TERC* against telomere content estimates as well as other markers. We also constructed linear models regressing merged telomere content as a function of various biomarkers with cell line primary site as a covariate. Moreover, we also calculated linear models of each of CCLE and GDSC telomere content as a function of *TERT* mRNA levels, *TERC* mRNA levels, primary site, TERTp mutant status, donor age, and calculated doubling time to examine the contributions of these specific factors. Out of these factors, we found *TERT* and *TERC* mRNA levels to be significantly correlated with both CCLE and GDSC telomere contents.

We also considered processed methylation estimates available on the CCLE data portal, namely the TSS 1 kb upstream estimates as well as the promoter CpG cluster estimates, which we correlated against telomere content estimates. For these annotations, we filtered out regions with a standard deviation of less than 0.05.

Results of *TERT* and *TERC* expression associations, as well as telomere content associations, are available in *Supplementary file 2*.

## Gene dependency associations

To identify gene dependencies associated with telomere content, we considered knockout/knockdown effects estimated in the Avana CRISPR-Cas9 (*Meyers et al., 2017*), Achilles RNAi

(*Tsherniak et al., 2017*), and DRIVE RNAi (*McDonald et al., 2017*) datasets. For Achilles and DRIVE, the datasets used were the April 2020 versions listed on the DepMap portal computed with DEME-TER2. For Avana, we used gene effect scores from the February 2021 release. For each gene dependency in each dataset, we computed the Pearson correlation coefficient against telomere content estimates generated separately with CCLE WGS and GDSC WES data. Correlation p values were determined using the two-tailed Student's *t* test. All correlation coefficients and p values were determined using the *pearsonr* function as part of the *scipy.stats* Python module.

To identify codependencies with the CST complex members, we employed an iterative approach to identify highly ranked correlations. In particular, starting with a seed set of genes (the base case of which was the CST complex), we searched for codependencies between two genes *x* and *y* under the criteria that the $r^2$ association between the two is among the top five for *x* vs. all other genes, and among the top five for *y* vs. all other genes as well. We recursively applied this method four times, which added the five shelterin components *ACD*, *POT1*, *TERF1*, *TERF2*, and *TINF2* to our gene set. To construct the clustered correlation matrix in *Figure 2c*, we used the *clustermap* function as provided by the Seaborn Python library, with Ward's method for the determination of the hierarchical clustering.

To identify significant associations between dependencies and mutations, we compared dependencies against binary categories of damaging/truncating (comprised of deleterious alterations, such as nonsense and splice-site alterations) and hotspot (highly recurrent) mutations. Using the previously downloaded DepMap 19Q4 mutation annotations, we considered mutations as 'damaging'/'truncating' if they were associated with a 'damaging' label under the *Variant_annotation* column, and we considered mutations as 'hotspot' if they were labeled as such in the corresponding COSMIC (*Tate et al., 2019*) (*isCOSMIChotspot*) or TCGA (*isTCGAhotspot*) columns. We excluded mutations with a total damaging or hotspot frequency of less than five across all profiled CCLE samples. Mutations were then compared with dependencies using a two-sided Mann-Whitney *U* test, with the two classes being non-damaging and non-hotspot mutant samples, and damaging and hotspot mutant samples, respectively.

To rank and visualize the codependencies shown in *Figure 2—figure supplement 2a,b* and the dependency-mutation associations shown in *Figure 2d,e*, we used a signed *q*-value approach. We first transformed the raw false discovery rates by taking the negative of the base-10 logarithm, and we then applied a sign to this transformed value as determined by the direction of the codependency (the sign of the correlation coefficient) or dependency-mutation association (negative for greater sensitivity in mutants, and positive otherwise).

Dependency analyses results are available in *Supplementary file 3*.

## Characterization of allele-specific TERT expression

Allele-specific expression may be detected by looking for discordant counts of reads mapping to single-nucleotide polymorphisms (SNPs) in DNA-sequencing vs. RNA-sequencing reads (*Huang et al., 2015*). In particular, allele-specific expression is evidenced by the biased frequency of a single allele of a heterozygous SNP in RNAseq reads compared to that of DNA-sequencing reads. To assess *TERT* expression in the context of allele-specificity, we examined cell lines for which DNA (WES or WGS) and RNA (RNAseq) sequencing data were available. To identify heterozygous anchor SNPs, we considered mutations in the *TERT* gene body called using Mutect 1.1.6 (*Cibulskis et al., 2013*) with default settings. We then applied a filter for mutations with at least eight reads supporting both the reference and alternate alleles that passed the Mutect quality control filter (i.e. classified as PASS). To force call the matching allele frequencies in RNA, we processed the matching aligned RNAseq reads using the ASEReadCounter tool provided in GATK 3.6 (*Van der Auwera et al., 2013*) with arguments -minDepth 8, –minBaseQuality 16, –minMappingQuality 255, and -U ALLOW_N_CIGAR_READS.

We then used these RNA and DNA allele frequencies to classify cell lines as monoallelic and biallelic expressors of *TERT* as well as two neighboring genes, *SLC6A19* and *CLPTM1L*. In particular, we examined the odds ratio derived from a binary contingency table with the two sets of categories being the context (DNA vs. RNA) and the allele (reference vs. alternate) of the read counts. To account for edge cases where the denominator of the odds ratio was zero, we added a pseudocount of 0.5 to each category before computing the odds ratio. We then denoted MAE lines as those having an odds ratio computed using the major allele as the denominator of greater than five. In

instances where there were multiple informative SNPs, we considered only the SNP with the greatest supporting total RNA-seq read count. In cases where the same SNP was detected across multiple sources (for instance, in both CCLE WES and WGS), we considered the source with the greatest coverage of the SNP.

Allele-specific calls for *TERT*, *SLC6A19*, and *CLPTM1L* are described in *Supplementary file 4*.

## Genome-wide allele-specific methylation analysis

To characterize and compare CpG-level ASM around the *TERT* genomic region, we utilized RRBS data generated by the CCLE (*Ghandi et al., 2019*). Mapped BAM files were downloaded from the CCLE FireCloud workspace, and ASM levels for each CpG pair were estimated using the *allelicmeth* command from the MethPipe package (*Song et al., 2013*). Within each sample, we first included only CpG pairs with a minimum coverage of eight reads. Next, among all 928 cell lines, CpG pairs included in less than 5% of these samples were excluded.

To estimate ASM, we employed a strategy similar to the original MethPipe ASM pipeline. For each pair of CpGs, we considered the four combinations of methylation states between the two CpGs: methylated-methylated (*mm*), methylated-unmethylated (*mu*), unmethylated-unmethylated (*uu*), and unmethylated-unmethylated (*mm*). For semi-methylated CpG pairs not subject to ASM, we would expect high and relatively equal frequencies of the *mu* and *um* pairs, whereas for ASM CpG pairs, we would expect the allele bias to result in high *mm* and *uu* counts and low *um* and *mu* counts. To quantify this imbalance, we used the mean square contingency coefficient ($\Phi$) with a pseudocount of 0.5. Namely, for each CpG pair, we computed

$$\Phi = \frac{\dot{mm} * \dot{uu} - \dot{mu} * \dot{um}}{\sqrt{\left(\dot{mm} + \dot{mu}\right)\left(\dot{um} + \dot{uu}\right)\left(\dot{mm} + \dot{um}\right)\left(\dot{mu} + \dot{uu}\right)}}$$

where $\dot{mm} = mm + 0.5$, $\dot{mu} = mu + 0.5$, $\dot{um} = um + 0.5$, and $\dot{uu} = uu + 0.5$. ASM CpG pairs therefore had a positive $\Phi$, whereas non-ASM pairs had a $\Phi$ of around 0. We rounded negative $\Phi$ values to 0. Before computing these imbalance values, we excluded CpG pairs with a methylation level of less than 0.1 or greater than 0.9 on either CpG, so as to filter out CpG pairs that were likely to be fully methylated or completely demethylated.

We first examined the ASM levels of the *TERT* locus, which considered as the *TERT* gene body as well as the flanking five kilobase regions. For these methylation estimates, we excluded CpGs with less than 25% valid ASM estimates. We segmented these CpGs into five regions: the promoter (chr5:1295246–1298643), CGI_1 (chr5:1294872–1295134), CGI_2 (chr5:1291374–1294439), CGI_3 (chr5: 1289695–1291090), and the remaining gene body (chr5: 1249661–1289359). We computed ASM imbalance values for these regions by taking the mean CpG-pair ASM values.

To identify genome-wide methylation events indicative of *TERT* promoter mutations, we searched for correlates with *TERT* promoter mutation status among average methylation levels of CpG islands (CGIs). CpG island annotations were downloaded from the UCSC genome browser at http://hgdownload.cse.ucsc.edu/goldenpath/hg19/database/cpgIslandExt.txt.gz. To filter out low-coverage CpG islands, we considered only CpG islands with at least eight CpG sites. Methylation levels per island were then estimated by taking the mean across all CpGs profiled within the island. Using the same filtering parameters, we also computed mean ASM estimates across these CGIs.

*TERT* locus methylation estimates are described in *Supplementary file 5*.

## Chromatin profiling data

To identify histone modifications associated with TERTp status, we downloaded global chromatin profiling data from the CCLE Data Portal (CCLE_GlobalChromatinProfiling_20181130.csv). Correlations between TERTp status and histone modification levels, as well as correlations between H3K9ac1K14ac0 levels and CGI ASM levels, are described in *Supplementary file 5*.

## Region enrichment analysis

To characterize the regions that were hypomethylated in association with TERT promoter mutations, we utilized Locus Overlap Analysis (LOLA) (*Nagraj et al., 2018*; *Sheffield and Bock, 2016*), which

discovers enriched regions among a background set. LOLA takes as input two region sets: the regions of interest and a background universe set. Both sets are then overlapped against a database of annotated regions, and overlapping and non-overlapping region frequencies are computed per annotated region set. Region overlap significance is then assessed against each annotated region set using Fisher's exact test with the two categories being the regions of interest vs. the background universe set and the overlap of each of these regions with the annotated region set.

We ranked hypomethylated CpG island regions by the significance of the change in TERTp wild type vs. TERTp mutant cell lines as assessed by a two-sided Mann-Whitney *U* test (i.e. regions with the most significant changes were ranked first). The top 1000 CpG islands were then used as the regions of interest, and the set of all CpG islands examined served as the background universe set. We utilized the LOLAweb application (at http://lolaweb.databio.org/), with the LOLACore and LOLARoadMap sets as the region databases.

Outputs of LOLA on CCLE *TERT* promoter-mutant hypomethylated CGIs are summarized in *Supplementary file 6*.

## TCGA data

TCGA methylation and normalized gene expression estimates were downloaded from the UCSC Xena browser (*Goldman et al., 2019*) (http://xena.ucsc.edu/, jhu-usc.edu_PANCAN_HumanMethylation450.betaValue_whitelisted.tsv.synapse_download_5096262.xena and EB++AdjustPANCAN_IlluminaHiSeq_RNASeqV2.geneExp.xena). Methylation levels of CGIs was estimated by averaging CpG methylation levels within each CGI, and CGIs with less than four profiled CpGs were excluded. *TERT* promoter mutation status was obtained from previous estimates (*Barthel et al., 2017*). The same hypomethylation-enrichment analysis previously described for the CCLE was then run on the top 1000 CGIs hypomethylated in *TERT* promoter-mutants using an identical ranking scheme.

A summary of LOLA results on TCGA methylation, *TERT* promoter status, and ALT-likelihood is provided in *Supplementary file 7*.

## Statistical analysis

Multiple hypothesis testing was accounted for using the Benjamini-Hochberg FDR with an alpha of 0.01 as provided by the *statsmodels* Python module.

# Acknowledgements

We thank Dr. Elizabeth Blackburn (UCSF) and Dr. Thomas Cech (University of Colorado Boulder) for providing insightful comments and feedback on the manuscript. We thank Dr. Hani Goodarzi (UCSF) for his generosity in providing storage and computing resources. We thank members of the Huang lab for their feedback.

# Additional information

## Competing interests

Mahmoud Ghandi: MG is an employee and holds equity in Monte Rosa Therapeutics and is a founding member at Cambridge Data Science LLC. The other authors declare that no competing interests exist.

## Funding

| Funder | Grant reference number | Author |
|---|---|---|
| Prostate Cancer Foundation | Young Investigator Award | Franklin W Huang |

The funders had no role in study design, data collection and interpretation, or the decision to submit the work for publication.

## Author contributions
Kevin Hu, Data curation, Software, Formal analysis, Investigation, Visualization, Methodology, Writing - original draft, Writing - review and editing; Mahmoud Ghandi, Conceptualization, Resources, Data curation, Software, Formal analysis, Supervision, Investigation, Visualization, Methodology, Project administration, Writing - review and editing; Franklin W Huang, Conceptualization, Resources, Supervision, Funding acquisition, Investigation, Methodology, Writing - original draft, Project administration, Writing - review and editing

## Author ORCIDs
Kevin Hu (iD) https://orcid.org/0000-0002-3631-8294
Mahmoud Ghandi (iD) https://orcid.org/0000-0003-1897-2265
Franklin W Huang (iD) https://orcid.org/0000-0001-5447-0436

## Decision letter and Author response
Decision letter https://doi.org/10.7554/eLife.66198.sa1
Author response https://doi.org/10.7554/eLife.66198.sa2

# Additional files
## Supplementary files
- Supplementary file 1. Telomere content estimates and sample information.
- Supplementary file 2. Telomere content and transcriptomic associations.
- Supplementary file 3. Unsupervised dependency associations.
- Supplementary file 4. Allele-specific expression calls.
- Supplementary file 5. *TERT* locus methylation.
- Supplementary file 6. Genome-wide methylation analysis in the CCLE.
- Supplementary file 7. Genome-wide methylation analysis in TCGA.
- Transparent reporting form

## Data availability
Telomere content estimates can be found in the supplementary materials and have been uploaded to the Cancer Dependency Map portal (https://depmap.org/portal/).

The following previously published datasets were used:

| Author(s) | Year | Dataset title | Dataset URL | Database and Identifier |
|---|---|---|---|---|
| Ghandi M, Huang FW, Jané-Valbuena J, Kryukov GV, Lo CC, McDonald ER, Barretina J, Gelfand ET, Bielski CM, Li H, Hu K, Andreev-Drakhlin AY, Kim J, Hess JM, Haas BJ, Aguet F, Weir BA, Rothberg MV, Paolella BR, Lawrence MS, Akbani R, Lu Y, Tiv HL, Gokhale PC, Weck S, MansourAA, Oh C, Shih J, Hadi K, Rosen Y, Bistline J, Venkatesan K, Reddy A, Sonkin D, Liu M, Lehar J, Korn JM, Porter | 2019 | Cancer Cell Line Encyclopedia | https://portals.broadinstitute.org/ccle | Broad Institute, ccle |

DA, Jones MD, Golji J, Caponigro G, Taylor JE, Dunning CM, Creech AL, Warren AC, McFarland JM, Zamanighomi M, Kauffmann A, Stransky N, Imielinski M, Maruvka YE, Cherniack AD, Tsherniak Z, Vazquez F, Jaffe JD, Lane AA, Weinstock DM, Johannessen CM, Morrissey MP, Stegmeier F, Schlegel R, Hahn WC, Getz G, Mills GB, Boehm JS, Golub TR, Garraway LA, Sellers WR

| Authors | Year | Title | URL | Source |
|---|---|---|---|---|
| Nusinow DP, Szpyt J, Ghandi M, Rose CM, McDonald ER, Kalocsay M, Jané-Valbuena J, Gelfand E, Schweppe DK, Jedrychowski M, Golji J, Porter DA, Rejtar T, Wang YK, Kryukov GV, Stegmeier F, Erickson BK, Garraway LA, Sellers WR, Gygi SP | 2020 | Quantitative Proteomics of the Cancer Cell Line Encyclopedia | https://gygi.hms.harvard.edu/publications/ccle.html | Gygi Lab, gygi.hms.harvard.edu/publications/ccle.html |
| DepMap | 2020 | DepMap 20Q2 Public | https://depmap.org/portal/download/ | Dep Map Portal, depmap.org/portal/download/ |
| Iorio F, Knijnenburg TA, Vis D, Bignell G, Menden M, Schubert M, Aben N, Gonçalves E, Barthorpe S, Lightfoot H, Cokelaer T, Greninger P, Chang H, Silva H, Heyn H, Deng X, Egan RK, Liu Q, Mironenko T, Mitropoulos X, Richardson L, Wang J, Zhang T, Moran S, Sayols S, Soleimani M, Tamborero T, López-Bigas N, Ross-MacDonald P, Esteller M, Gray N, Haber D, Stratton MR, Benes C, Wessels L, Saez-Rodriguez J, McDermott U, Garnett M | 2016 | GDSC whole-exome sequencing data | https://www.ebi.ac.uk/ega/studies/EGAS00001000978 | European Genome-phenome Archive, EGAS00001000978 |

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
