## [Decision Letter]

**Acceptance summary:**

This manuscript is of interest for researchers in the cancer genomics and telomere maintenance fields. Telomere maintenance is critical in tumour development as it allows cells to divide indefinitely. By studying a large collection of cell lines, the authors have identified genomic features broadly correlated with telomere content, including telomerase expression and mutation, as well as differences between tissues of origin, dependencies among different telomere maintenance genes, and correlations with region-specific methylation patterns. Overall, its value lies in the novel biological insights it provides regarding telomere maintenance and the massive resource it provides to the scientific community.

**Decision letter after peer review:**

Thank you for submitting your article "Integrated evaluation of telomerase activation and telomere maintenance across cancer cell lines" for consideration by *eLife*. Your article has been reviewed by 3 peer reviewers, including C Daniela Robles-Espinoza as the Reviewing Editor and Reviewer #1, and the evaluation has been overseen by Maureen Murphy as the Senior Editor. The following individual involved in review of your submission has agreed to reveal their identity: Floris Barthel (Reviewer #2).

In this manuscript, Hu, Ghandi and Huang describe a number of genomic features that are correlated with telomere content, and focus on the study of cancer cell lines from two different sources (the Cancer Cell Line Encyclopedia, CCLE) and the Genomics of Drug Sensitivity in Cancer (GDSC). They describe genomic alterations, gene dependencies, telomerase (TERT) expression and promoter mutation, and methylation patterns both in the TERT locus and beyond. The study is impressive in scope, as putting together data from distinct cell lines from different batches and sources is not an easy task.

However, the reviewers have a number of suggestions to improve the conclusions and readability of the manuscript.

Essential revisions

1. About the structure of the manuscript.

Two reviewers commented that the manuscript is at times hard to understand due to it covering a large number of observations that not always follow logically from each other. Specifically, it was mentioned that the ATRX/DAXX section does not add much to the paper, and that a substantial part of analyses and supplementary data are not mentioned (e.g., What is CERES and DEMETER2 dependency, in Figure S6c and d?). Please re-check these issues, and if possible, simplify the paper to convey a clear message.

2. About cell lines.

Please answer the following questions.

a. Does the CCLE provide any information on cell line source? Do the authors have any cell lines with biological replicates from different sources that could determine how this affects telomere length, as in eg. PMID 30089904? Also, are there any technical replicates derived from unique libraries prepared from the same cell line to establish that these show similar measurements?

b. Do the authors think their conclusions are affected by the fact that cell lines are immortalised, and this process in many cases includes the activation of TERT? Can their conclusions be extended to living tissues?

c. Can the number of cell lines from both sets that were taken forward for downstream analyses be specified in the main text?

3. About telomere measurements and correlations.

a. The raw telomere content measurements from WGS are not discussed much and quickly discarded in favor of normalized estimates. Looking at the supplementary table these look to be all on a similar and comparable scale for WGS. It strikes me that WGS-based estimates are much more reliable, and some associations may be stronger using these numbers despite being limited to a much smaller sample size. Can the authors show the overlap between WGS/WXS and associated data (CRISPR screens, RNAseq, etc)? Do the described findings hold when limited to WGS raw estimates or is there insufficient power? Also, an explanation of why correlation is higher between WES and WGS of different datasets than with those of the same dataset would be welcome.

b. While the authors look at the patients age at tissue extraction and find a correlation between estimated telomere content and age, one wonders if cell culture specific parameters could not be even stronger determinants. Have the authors looked at time in culture, passage number and population doubling level (PDL) in relation to telomere length?

c. Could the authors infer ALT status for a large number of cell lines from other resources? Capital Biosciences was awarded NIH SBIR funding several years ago specifically to determine the ALT status of all cell lines in the ATCC.

See for example:

1. https://www.eposters.net/pdfs/identification-of-new-alternative-lengthening-of-telomeres-positive-cancer-cell-lines-using-the-c.pdf

2. https://www.atcc.org/~/media/PDFs/Presentations/HOC%20poster%20ALT.ashx

At a minimum, perhaps the authors could annotate lines known to exhibit ALT as ALT+ and others as ALT unknown?

d. It is slightly surprising there is a positive correlation between telomere content and TERT mRNA in CNS since CNS tumors are often ALT+ (and telomerase-). Possibly none of the CNS lines assayed were ALT+? Perhaps the correlation between TERT mRNA and telomere content needs to be adjusted for ALT status because of the presumed interaction. Could the authors incorporate multiple correlates of telomere content in a model and determine their independent contributions and interactions? Perhaps this could be limited to WGS and using the raw measurements.

e. Some correlations with TC (for example all Figure S3) are likely to be biased by the important TC differences between tissue types. Can the authors perform this analysis on each tissue separately, or normalize TC to use relative TC within a tissue type?

4. About figures.

a. Figure 1.

– This is an informative figure. Can a clarification of the total number of cell lines be added?

– It says it is 683 but from the main text it seems that many more were taken forward for analysis (1,056 GDSC with WES and 329 WGS CCLE, with only 286 overlapping)? Why are the others not depicted here?

– If the outlier U2-OS cell line is removed from the 44 bone cell lines, is the bone cell line average still the highest?

– Can the ATRX/DAXX expression be included?

b. Figure 2.

– Figure 2a-b: the authors claim that dependency toward TRF1 and the CST complex is higher in cells with short telomeres. However, they show a positive correlation between TC and AVANA dependency. This may be interpreted that cells with longer telomeres are more sensitive to loss of TRF1 or the CST complex. That could be explained by their role during telomere replication, and the likely higher replication stress at longer telomeres. If this is not the correct explanation, can the explanation be reworded, as it is confusing?

– Figure 2d-e: Does each dot represent a cell line? If so, are the conclusions (model in 2f) drawn out of only 1 cell with a mutation in a hotspot?

c. Figure 3a.

– Here, it seems that cells with lower PML expression are more sensitive to DAXX suppression. Could the analysis be done with telomerase positive cells only? Similarly, cells with higher levels of ZMYM3 are more sensitive to DAXX suppression. What does that mean biologically? Can you please specify what are the dots?, do they refer to cell lines? So these all represent the top score of their corresponding cell line?

5. Other points

a. The introduction and the narrative tell us that telomerase is activated, that telomeres are usually longer in cancer, etc. Therefore, the phrase "The 55 non-cancerous samples profiled as part of the GDSC also displayed relatively high telomere contents (P = 5.9×10-4, two sided Mann-Whitney U test), consistent with previous reports of widespread telomere shortening in cancer (Barthel et al., 2017)." seems to contradict what has been said. So, is it expected that the normal samples would have long telomeres? Would this be expected physiologically or is it an effect from these being cell lines? Also – is the information for these normal cell lines displayed in Figure 1? Reviewers recommend to add the normals to this figure so it is easier to compare between normal and cancer tissues.

b. Could the definition of what a dependency means, exactly, be included? Does it assess correlated gene expression? Does it assess activation of one gene upon knockout of the other? It would be helpful to spell this out in the main text. This may help understand the observed dependencies between the members of the shelterin complex and those of the CST complex.

c. Page 6 says "increased sensitivities to knockout of the CST complex components, which are key mediators of telomere capping and elongation termination (Chen et al., 2012), were correlated with lower telomere content." However, it seems that supplementary figure 3b indicates that the CST genes are positively correlated with telomere content? If this is an index of sensitivity, could this be indicated somewhere? (At the moment it seems similar to the TERT figure, so readers may read it in the same way which may be confusing). Could TERF1 be highlighted in Figure S3b please, DRIVE dependencies?

d. About methylation studies. In page 11 it says "TERTp mutants exhibited strong and significant (P < 0.01) increases in ASM (allele-specific CpG methylation) in the promoter (n = 485), remaining gene body (n = 493), and exon 1 (n = 478) regions", but then in page 12, line 292 it says "The observation that TERT promoter mutants display a hypomethylated TERT locus…" – does this not directly contradict the statement above? Or is it referring to methylation on a specific region? Could this be clarified please?

*Reviewer #1:*

In this manuscript, Hu, Ghandi and Huang describe a number of genomic features that are correlated with telomere content, and focus on the study of cancer cell lines from two different sources (the Cancer Cell Line Encyclopedia, CCLE) and the Genomics of Drug Sensitivity in Cancer (GDSC). They describe genomic alterations, gene dependencies, telomerase (TERT) expression and promoter mutation, and methylation patterns both in the TERT locus and beyond. The analyses are carefully done and largely convincing, and impressive in scope as putting together data from distinct cell lines from different batches and sources is not an easy task.

Their major claims are that:

– TERT expression correlates with telomere content in lung, central nervous system, and leukemia cell lines. This claim is supported by their analyses of telomere tract quantification and TERT mRNA levels in the same cell lines (Figure S4). However, some negative correlations in other cell lines were also found, which are currently unexplained.

– Lower telomeric content is associated with dependency of CST telomere maintenance components. This is supported by an analysis of the Avana dependency dataset, which has been previously published. This makes sense biologically and is clear in Figure S3.

– Increased dependencies of shelterin member genes are associated with wild-type TP53 status. An extended analysis as the one before in the Avana and DRIVE dependency datasets support this claim. This would mean that cells are less sensitive to shelterin depletion when they are mutated in TP53, which makes sense biologically.

– Monoallelic expression in TERT promoter-mutant contexts. The analysis behind this claim, based on alignment of RNA reads to heterozygous regions and the subsequent counting of reads mapping to each allele, seems robust, although it would have been nice to see a few more details (for example, what is the measure of linkage disequilibrium between the TERT promoter mutations and these anchor SNPs).

– TERT promoter-mutant cell lines show hypomethylation at PRC2-repressed regions. This claim is based on an analysis of genome-wide reduced representation bisulfite sequencing (RRBS) data in nearly 1000 cell lines, and the finding of associations between TERT promoter mutation and CpG methylation of different regions of the TERT gene and other loci. An overlap analysis showed that these methylated regions were enriched for PRC2-repressed regions.

All in all, I believe the claims in this study are well supported. This work constitutes a valuable resource for the scientific community, and has made new observations that contribute to the elucidation of telomere maintenance mechanisms.*Reviewer #2:*

Kevin Hu, Mahmoud Ghandi and Franklin W. Huang present their important work on determining telomere content across > 1000 cancer cell lines. In addition to this highly valuable dataset, the authors present several scientific vignettes wherein various questions are addressed using their curated resource. Using CRISPR/Cas9 screening data from the same set of cell lines, they find that sensitivity to CST (CTC1, STN1, TEN1) knockout was associated with telomere content and that cell lines sensitive to CST knockout demonstrated lower telomere content. Shelterin complex members (ACD, POT1, TERF1, TERF2, TINF2) were further identified as co-dependencies to CST knockout.

They find that cell lines with monoallelic expression, as in a TERT promoter mutant setting, also demonstrate allele-specific methylation. Interestingly, TERT promoter mutant demonstrated gene body hypomethylation not observed in other monoallelic TERT expressors. Moreover, this hypomethylation was found to be genome-wide and not restricted to the TERT locus. Finally, nearly all of this hypomethylation was localized in PRC2-repressed regions.

The hypothesis-driven vignettes are clearly of interest in providing correlative insights that could fuel future mechanistic studies. More importantly, they exemplify the strengths of the resource. Nevertheless, the primary value of the manuscript in my opinion is the enormous resource of telomere content estimates for cell lines widely used in biomedical research. The authors can do a more thorough job showing the reader that their measurements are reliable and whether or not they would be generally applicable to fresh cell lines purchased from a vendor.

1. WGS-based telomere content estimates are likely much more reliable than WXS-based estimates. The authors do not comment on this.

2. Cell lines from different sources can vary substantially. For example, HeLa strains from different laboratories can show vastly different karyotypes (PMID 30778230). These caveats are not currently discussed.

3. Telomere length attrition is replication dependent, however the authors have not looked at whether their measurements are associated with population doublings in individual samples or across multiple samples taken at different PDLs.

4. ALT status is an important confounder of telomere content. The authors hint at ALT status in their manuscript at various points, but do not incorporate ALT status into their analyses.

5. The combination of a number of parameters is going to determine telomere content in cell lines. It would be helpful to understand the combined contribution of multiple parameters.*Reviewer #3:*

In this manuscript, Huang and colleagues used public whole genome or whole exome sequencing datasets to establish telomere content in over a thousand cancer cells lines. Once telomere content was established, they linked it to other available sets of analysis, including RNAseq libraries and genome-wide CRISPR or RNAi screens.

First, they found that telomere content is highly variable among tissue types, with highest telomere content (TC) in tissues that have higher frequencies of ALT activation. They also found weak correlation with TERT or TERC expression. They then analyzed the correlation between telomere content and gene dependencies and found that dependencies to TRF1 and the CST correlates with TC. Independently of TC, they analyzed correlations between gene dependency of telomeric proteins or ATRX and DAXX with mutations or transcriptomic and proteomic profiles. Finally, they analyzed the link between TERT allelic expression and epigenetic marks on TERT promoter.

Although such broad analysis of telomere length or telomere proteins / TERT dependencies is interesting, it appears hard to really see the biological significance of certain correlations or the novelty of the findings.

---

## [Author Response]

Essential revisions1. About the structure of the manuscript.Two reviewers commented that the manuscript is at times hard to understand due to it covering a large number of observations that not always follow logically from each other. Specifically, it was mentioned that the ATRX/DAXX section does not add much to the paper, and that a substantial part of analyses and supplementary data are not mentioned (e.g., What is CERES and DEMETER2 dependency, in Figure S6c and d?). Please re-check these issues, and if possible, simplify the paper to convey a clear message.

Thank you for this comment and we agree. To improve the flow and clarity of the paper we have now removed the ATRX/DAXX section from the paper. We have also revised the axis labels of Figure S6c and d such that CERES and DEMETER2 dependency are replaced by Avana and DRIVE (the correct datasets used, whereas CERES and DEMETER2 correspond to the scoring algorithms). We appreciate the opportunity to simplify the paper.

2. About cell lines.Please answer the following questions.a. Does the CCLE provide any information on cell line source?

Yes, the CCLE does provide additional cell line source attributes. We have added these sources, as well as additional information on growth medium, freezing medium, doubling time, pathologist’s annotation, and TCGA code to Supplementary Table S1.

Do the authors have any cell lines with biological replicates from different sources that could determine how this affects telomere length, as in eg. PMID 30089904?

The CCLE WES, CCLE WGS, and GDSC WES datasets are of different biological replicates, and we show that there is a moderate correlation between CCLE WES and CCLE WGS telomere content as well a strong correlation between CCLE WGS and GDSC WES telomere content.

Also, are there any technical replicates derived from unique libraries prepared from the same cell line to establish that these show similar measurements?

We thank the reviewers for raising this question. Within the CCLE WGS dataset, we have since determined that the HEL 92.1.7 cell line was twice-sequenced. We have now run Telseq on this second sequencing run and found both estimates to be about 4 kilobases, showing a similar measurement. We have updated the manuscript accordingly.

b. Do the authors think their conclusions are affected by the fact that cell lines are immortalised, and this process in many cases includes the activation of TERT?

As cell lines are immortalised frequently through the activation of TERT, we think this is a useful setting in which to study telomerase expression and telomere length. So while we cannot exclude the fact that this common process is affected differently in cancer tissues, we think the immortalization process in cell lines offers a reasonable comparison to cancers and think our conclusions can also apply to cancer tissues.

Can their conclusions be extended to living tissues?

This is a very interesting question. We think that while there certainly are limitations of cancer cell lines in terms of their similarity to tumor tissues, the extent to which they can be studied and perturbed affords significant advantages, and common next-gen sequencing methods can be used alongside to assess telomere content, since many advanced cancers are now undergoing next-gen sequencing as part of routine clinical care. We do think our conclusions about the sensitivities of cancer cell lines based on telomere content to depletion of telomere maintenance mechanisms opens potential new strategies for targeting cancers by taking into account telomere content/length that will need further validation.

c. Can the number of cell lines from both sets that were taken forward for downstream analyses be specified in the main text?

Yes, we have now specified this number in the main text.

3. About telomere measurements and correlations.a. The raw telomere content measurements from WGS are not discussed much and quickly discarded in favor of normalized estimates. Looking at the supplementary table these look to be all on a similar and comparable scale for WGS.

Although raw telomere content measurements from WGS are reliable based on the results of the original Telseq paper, we ultimately used a z-scored transformation to make these estimates more compatible with the WES-based content estimates. In addition, we applied a log-transformation on all telomere content estimates because of a strong skew towards cell lines with markedly high telomere contents.

To illustrate that our transformations do not markedly affect the results, we have added the raw CCLE WGS telomere content estimates to Figure S5a (correlation plots of CST complex dependencies versus telomere content). In these examples, we show that raw estimates are still correlated with the dependencies, although the linear relationship is weaker due to the skew in the raw telomere content.

It strikes me that WGS-based estimates are much more reliable, and some associations may be stronger using these numbers despite being limited to a much smaller sample size.

We agree with this point, and we would like to note that we indeed ran analyses such as the associations between telomere content and dependency (Figure 2), between telomere content and age as well as tissue type (Figure S2), and between telomere content and TERT and TERC transcriptomics. For Figure S3, we decided against using WGS estimates along because the overlap between WGS and datasets such as mass-spectrometry protein expression and DRIVE dependencies would have been relatively small, and because we believed that any strong associations would have been present anyway, as with the Avana associations with TERF1 and the CST complex members.

Can the authors show the overlap between WGS/WXS and associated data (CRISPR screens, RNAseq, etc)?

Yes, we have now added a summary figure to Figure S1 showing the totals within each dataset as well as the number of individual cell lines containing each annotation, in the format of a binary heatmap.

Do the described findings hold when limited to WGS raw estimates or is there insufficient power?

The described findings hold when limited to WGS raw estimates. The transformation used to produce the WGS transformed estimates was just a log-transformation followed by a linear z-score scaling, so our findings are not so sensitive to whether the WGS raw or transformed estimates were used. To illustrate, we have added plots of raw CCLE WGS telomere content versus CST complex dependencies to Figure S5.

Also, an explanation of why correlation is higher between WES and WGS of different datasets than with those of the same dataset would be welcome.

We have added histograms showing the distribution of the total numbers of reads containing at least 6 telomeric repeats among cell lines for each sequencing dataset to Figure S1. CCLE WGS lines had around 100,000 such reads, whereas GDSC WES lines had about 100-1,000 and most CCLE WES lines had less than 100. These read numbers suggest that CCLE WGS offers the most accurate estimates, followed by GDSC WES and then CCLE WES. Moreover, the difference in total telomeric reads detected would also explain why the correlation between CCLE WGS and GDSC WES (r=0.84) is much higher than that between CCLE WGS and CCLE WES (r=0.71), which itself exceeds that of CCLE WES versus GDSC WES (r=0.60).

b. While the authors look at the patients age at tissue extraction and find a correlation between estimated telomere content and age, one wonders if cell culture specific parameters could not be even stronger determinants. Have the authors looked at time in culture, passage number and population doubling level (PDL) in relation to telomere length?

We thank the Reviewer for this suggestion. We were able to find annotations for calculated doubling time in the CCLE, which appear to be weakly negatively correlated with telomere content. We have appended these plots to Figure S2.

c. Could the authors infer ALT status for a large number of cell lines from other resources? Capital Biosciences was awarded NIH SBIR funding several years ago specifically to determine the ALT status of all cell lines in the ATCC.See for example:1. https://www.eposters.net/pdfs/identification-of-new-alternative-lengthening-of-telomeres-positive-cancer-cell-lines-using-the-c.pdf2. https://www.atcc.org/~/media/PDFs/Presentations/HOC%20poster%20ALT.ashxAt a minimum, perhaps the authors could annotate lines known to exhibit ALT as ALT+ and others as ALT unknown?

We have compiled a table of ALT status across the linked studies as well as additional ones from the literature and appended these to Table S1, with 76 annotated cell lines in total.

d. It is slightly surprising there is a positive correlation between telomere content and TERT mRNA in CNS since CNS tumors are often ALT+ (and telomerase-). Possibly none of the CNS lines assayed were ALT+? Perhaps the correlation between TERT mRNA and telomere content needs to be adjusted for ALT status because of the presumed interaction. Could the authors incorporate multiple correlates of telomere content in a model and determine their independent contributions and interactions? Perhaps this could be limited to WGS and using the raw measurements.

We have constructed models incorporating TERT mRNA expression, TERC mRNA expression, cell line primary site, TERTp status, donor age, and calculated doubling time for predicting CCLE WGS and GDSC WES telomere content separately. The outputs of these models are detailed in Table S2.

Of the CNS lines for which we found ALT status in the literature, we know that PFSK1 and DAOY are ALT+, and we also know that A172, KNS42, U251MG, MOGGUVW, and SW1088 are not. The majority of CNS tumors in this dataset from the CCLE are TERT+.

e. Some correlations with TC (for example all Figure S3) are likely to be biased by the important TC differences between tissue types. Can the authors perform this analysis on each tissue separately, or normalize TC to use relative TC within a tissue type?

This is an interesting suggestion. We have updated the analysis to include tissue type as a covariate (Table S2 and Figure S4b).

4. About figures.a. Figure 1.– This is an informative figure. Can a clarification of the total number of cell lines be added?– It says it is 683 but from the main text it seems that many more were taken forward for analysis (1,056 GDSC with WES and 329 WGS CCLE, with only 286 overlapping)? Why are the others not depicted here?

We thank the reviewer for this positive feedback. In this figure, we omitted cell lines that lack expression and copy number annotations for TERT and TERC, merged (CCLE WGS and GDSC WES) telomere content estimates, and ATRX and DAXX mutation profiling for clarity. We also omitted tissue types with less than 10 cell lines present after this preliminary filtering was applied. This filtering was done for clarity, as it would have been difficult to display cell lines with missing values in a concise format. (We have since made an exception for non-cancerous cell lines as per the recent comments). In total, 738 cell lines are shown in this figure.

– If the outlier U2-OS cell line is removed from the 44 bone cell lines, is the bone cell line average still the highest?

If the U2-OS cell line is removed, the bone cell line average is no longer the highest (blood becomes the highest). To account for the impact of cell lines with particularly high telomere contents, we have revised this figure to sort by median, rather than mean, telomere content.

– Can the ATRX/DAXX expression be included?

We have now added ATRX and DAXX expression levels to the bottom of the Figure 1.

b. Figure 2.– Figure 2a-b: the authors claim that dependency toward TRF1 and the CST complex is higher in cells with short telomeres. However, they show a positive correlation between TC and AVANA dependency. This may be interpreted that cells with longer telomeres are more sensitive to loss of TRF1 or the CST complex. That could be explained by their role during telomere replication, and the likely higher replication stress at longer telomeres. If this is not the correct explanation, can the explanation be reworded, as it is confusing?

We thank the Reviewer for this helpful suggestion. We have reworded our discussion of the association between telomere content and CST complex dependencies to make this point clearer. In particular, the numerical dependency scores that we use are calculated such that a lower score indicates a stronger (or higher) dependency. Therefore, we meant to say that cells with longer telomeres are less sensitive to loss of TRF1 and the CST complex, as evidenced by a higher raw dependency score.

– Figure 2d-e: Does each dot represent a cell line? If so, are the conclusions (model in 2f) drawn out of only 1 cell with a mutation in a hotspot?

Here, each dot represents a mutation, and we aim to show the mutations that are associated with the respective dependencies of telomere-related proteins. We have remade this figure in the format of a volcano plot to make this point more clear.

c. Figure 3a.– Here, it seems that cells with lower PML expression are more sensitive to DAXX suppression. Could the analysis be done with telomerase positive cells only? Similarly, cells with higher levels of ZMYM3 are more sensitive to DAXX suppression. What does that mean biologically? Can you please specify what are the dots?, do they refer to cell lines? So these all represent the top score of their corresponding cell line?

In response to previous concerns regarding the low contribution of this figure to the paper, we have removed this figure from the text.

5. Other pointsa. The introduction and the narrative tell us that telomerase is activated, that telomeres are usually longer in cancer, etc. Therefore, the phrase "The 55 non-cancerous samples profiled as part of the GDSC also displayed relatively high telomere contents (P = 5.9×10-4, two sided Mann-Whitney U test), consistent with previous reports of widespread telomere shortening in cancer (Barthel et al., 2017)." seems to contradict what has been said. So, is it expected that the normal samples would have long telomeres? Would this be expected physiologically or is it an effect from these being cell lines? Also – is the information for these normal cell lines displayed in Figure 1? Reviewers recommend to add the normals to this figure so it is easier to compare between normal and cancer tissues.

We appreciate this comment and agree it enables a better comparison and have now added the normal cell lines to this figure (they were previously excluded due to a lack of expression and copy number profiling in the CCLE). We have also revised the introduction to state that cancers in general tend to have shorter telomeres than normal tissues, perhaps as a result of telomere maintenance mechanisms emerging only after telomeres have become sufficiently short in a crisis or because there is some fitness advantage to shorter telomeres.

b. Could the definition of what a dependency means, exactly, be included? Does it assess correlated gene expression? Does it assess activation of one gene upon knockout of the other? It would be helpful to spell this out in the main text. This may help understand the observed dependencies between the members of the shelterin complex and those of the CST complex.

We thank the Reviewer for this question. We have now added an explanation of the numerical value of a gene dependency to the “Telomere content associates with CST complex dependencies” section. In particular, a dependency score reflects how sensitive a particular cell line is to the inactivation of a particular gene, with more negative scores reflecting increased sensitivity.

c. Page 6 says "increased sensitivities to knockout of the CST complex components, which are key mediators of telomere capping and elongation termination (Chen et al., 2012), were correlated with lower telomere content." However, it seems that supplementary figure 3b indicates that the CST genes are positively correlated with telomere content? If this is an index of sensitivity, could this be indicated somewhere? (At the moment it seems similar to the TERT figure, so readers may read it in the same way which may be confusing). Could TERF1 be highlighted in Figure S3b please, DRIVE dependencies?

Consistent with the above comment, we have revised our wording to clarify our results here. A negative dependency score reflects increased sensitivity, so a positive correlation between telomere content and dependency scores suggests that cell lines with higher telomere content are less sensitive to inactivation of the CST complex members.

We have now highlighted TERF1 in Figure S3b.

d. About methylation studies. In page 11 it says "TERTp mutants exhibited strong and significant (P < 0.01) increases in ASM (allele-specific CpG methylation) in the promoter (n = 485), remaining gene body (n = 493), and exon 1 (n = 478) regions", but then in page 12, line 292 it says "The observation that TERT promoter mutants display a hypomethylated TERT locus…" – does this not directly contradict the statement above? Or is it referring to methylation on a specific region? Could this be clarified please?

We appreciate the opportunity to make this clarification. We state that TERTp mutants exhibited increases in ASM (allele-specific CpG methylation) yet show a hypomethylated TERT locus. This statement is consistent, because the increase in ASM is due to the hypomethylation (loss of methylation) at the TERT locus on a single allele. We have reworded this section to make this point clearer to the reader.